# Multidrug-Resistant *Acinetobacter baumannii* in Jordan

**DOI:** 10.3390/microorganisms10050849

**Published:** 2022-04-20

**Authors:** Mohammad Al-Tamimi, Hadeel Albalawi, Mohamd Alkhawaldeh, Abdullah Alazzam, Hassan Ramadan, Majd Altalalwah, Ahmad Alma’aitah, Dua’a Al Balawi, Sharif Shalabi, Jumana Abu-Raideh, Ashraf I. Khasawneh, Farah Alhaj, Kamal Hijawi

**Affiliations:** 1Department of Basic Medical Sciences, Faculty of Medicine, The Hashemite University, Zarqa 13133, Jordan; hadeelkhaleel100@yahoo.com (H.A.); 1636786@std.hu.edu.jo (M.A.); 1637411@std.hu.edu.jo (A.A.); 1635089@std.hu.edu.jo (H.R.); 1635126@std.hu.edu.jo (M.A.); 1637475@std.hu.edu.jo (A.A.); albliwidoaaibrahim@gmail.com (D.A.B.); smshalabi459@gmail.com (S.S.); salmajumanasami@gmail.com (J.A.-R.); ashrafkh@hu.edu.jo (A.I.K.); f.alhaj1@yahoo.com (F.A.); 2Department of Medical Laboratory Sciences, Faculty of Allied Health Sciences, The Hashemite University, Zarqa 13133, Jordan; kamal-hijjawi@outlook.com

**Keywords:** multi-drug resistance (MDR), *Acinetobacter baumannii*, Jordan, carbapenemase, OXA, VIM, IMP, KPC

## Abstract

Background: *Acinetobacter baumannii* is a common cause of multi-drug (MDR)-resistant infections worldwide. The epidemiological and molecular characteristics of MDR-*A. baumannii* in Jordan is not known. Methods: *A. baumannii* isolates were collected from 2010 to 2020 from three tertiary hospitals in Jordan. Demographic and clinical data, isolates information, antibiotic susceptibility patterns, phenotypic, and molecular characterization of carbapenem resistance genes were performed. Results: A total of 622 *A. baumannii* isolates were collected during the study period. Most isolates were from males, aged 18–60 years, Jordanian, from infected wounds, and were patients in surgery or critical care units. Among patients from whom *A. baumannii* was isolated, associated risk factors for MDR were adults over 60, males, critically ill patients and infected wounds (OR 4.14, 2.45, 10, 7, respectively, *p* < 0.0001). Incidence rates from 2010 to 2015 showed a slight increase in MDR (3.75/1000 to 4.46/1000). Resistance patterns indicated high resistance for most cephalosporins, carbapenems, and fluoroquinolones, moderate resistance for trimethoprim/sulfamethoxazole and ampicillin/sulbactam, low resistance for aminoglycosides and tetracyclines, while colistin and tigecycline, have the lowest resistance rates. 76.8% of *A. baumannii* isolates were MDR and 99.2% were carbapenem-resistant. All isolates were positive for the OXA-51 gene (100%), 98.5% were positive for the OXA-23 gene, 26.6% for the VIM gene, while KPC and IMP genes were almost not detected (0% and 0.8% respectively). Conclusions: This is the first large, multicentric, prolonged study that provides insights into *A. baumannii* infections in Jordan. Attention to patients at higher risk is important for early identification. Colistin and tigecycline were the most effective antimicrobials.

## 1. Introduction

*Acinetobacter* is an aerobic, Gram-negative bacillus, pleomorphic and non-motile bacterium. It is common among immunocompromised individuals, particularly those who have a prolonged hospital stay in an intensive care setting [1,2]. *Acinetobacter* genus now consists of 26 named species and nine genomic species. Four species of Acinetobacters (*A. calcoaceticus*, *A. baumannii*, *Acinetobacter* genomic species 3 and *Acinetobacter* genomic species 13TU) are difficult to differentiate, and are often referred to as the *A. calcoaceticus*-complex. Although *A. calcoaceticus* has not been implicated in clinical disease, the other three species in the *A. calcoaceticus*-complex are the most clinically significant species, with *A. baumannii* being the most important and relevant to drug-resistant infections [2,3].

*A. baumannii* is commonly associated with bacteraemia, pulmonary infections, meningitis, battlefield wound infections and urinary tract infections [2,4]. The risk of infection increases dramatically when medical devices such as ventilators, endotracheal incubators, and catheters are used [2,5]. The mortality rates of *A. baumannii*-related infections can reach 35% and above [1,5,6]. Recent increases in incidence of *A. baumannii* infection were observed in infected combat troops returning from conflict zones including Iraq and Afghanistan [2,7].

*A. baumannii* is one of the six most important multi-drug-resistant microorganisms (MDR) in hospitals worldwide [5]. *A. baumannii* resistance rates vary from country to country but have been increasing over time. Reported rates of resistance to gentamycin and ceftazidime ranged from 0 to 81%, amikacin resistance ranged from 10 to 51%, ciprofloxacin resistance ranged from 19 to 81%, while piperacillin-tazobactam resistance rates ranged from 36 to 75% depending on the country [5]. The most effective drugs were imipenem and meropenem against *A. baumannii* infections. However, recent studies have shown the emergence of resistance rates as high as 87% according to clinical setting and geographic distribution with these antimicrobials, leaving colistin or tigecycline as the only available treatments for MDR *A. baumannii* infections. Unfortunately, resistance to colistin and/or tigecycline has recently emerged in Europe with an incidence rate of 3–6% [3]. Increased incidence of *Acinetobacter* species resistant to almost every available anti-microbial agent is of great concern [3].

Data available on MDR *Acinetobacter* species in the Middle East region are limited [8]. In Saudi Arabia, 191 confirmed isolates of *A. baumannii*, from blood, had high resistance to most antimicrobials including imipenem (61.3%) and meropenem (60.7%) but low rates of resistance to colistin (0.5%) and tigecycline (3.9%). MDR was observed in 69% of the *A. baumannii* isolates and 78.8% of these MDR strains showed sensitivity only to colistin and tigecycline [9]. In Iran, 221 clinical isolates and 22 environmental *A. baumannii* isolates showed resistance to the majority of antimicrobials tested ranging from 69% to 100%, with the exception of tigecycline and colistin [10]. Two studies in Jordan using a limited number of patients investigated the rate of *A. baumannii* isolates in different patient’s populations and concluded the MDR rate was in the range of 65.7% and 77.6% [11,12]. Other Middle Eastern studies investigated *A. baumannii* infection rates in critically ill patients and concluded that the rate of MDR was as high as 100% [13,14,15,16].

To the best of our knowledge, there is no updated information regarding *A. baumannii* infection incidence, antibiotic susceptibility pattern, and antibiotic resistance genes in Jordan. Thus, the aims of this study were to evaluate the incidence rate of multi-drug-resistant *A. baumannii* infection, phenotypic and genotypic characterization of carbapenem resistance genes using large number of isolates, collected over a long period and from different centers and clinical setting in Jordan.

## 2. Materials and Methods

### 2.1. A. baumannii Isolates Collection

Bacterial isolates (consecutive and non-duplicated) were collected from January 2010 to June 2020 from three central tertiary hospitals located in the capital city of Amman (The Specialty hospital, Prince Hamzah hospital, and the Islamic hospital) after obtaining formal patient consent. Important demographic, clinical and laboratory information were reordered, and samples were obtained and stored for further analysis. Various specimens were studied including urine, respiratory samples (sputum and broncho alveolar lavage), blood, wound pus, body fluids and tissue. Data including patient demographics (age, gender, residency and nationality), date of admission and discharge, hospital department, specimen type, test date, microorganism identified with confidence rate, and antibiotic susceptibility pattern were collected for each patient from Vitek 2 system database and patients files. The study protocol was approved by the institutional review board (IRB) of the Hashemite University and IRB of relevant hospitals.

### 2.2. Bacterial Identification

Specimens were plated on general and selective media, including Mac-Conkey agar, blood agar, and CHROM agar followed by Gram stain, manual and automated biochemical tests. Isolates identities were further analyzed using Vitek 2 Compact System with Gram-negative detection kit (BioMérieux, Marcy l’Etoile, France). The identity of the isolates was confirmed by conventional PCR to detect the presence of the intrinsic *bla_OXA-51_*-like gene, using specific primers as previously described [17]. Samples that were *bla_OXA-51_*-negative were excluded from the study.

### 2.3. Antimicrobial Susceptibility Testing

Antibiotic susceptibility pattern was performed using Vitek 2 system with Gram-negative susceptibility card GN-69 (BioMérieux, Marcy l’Etoile, France). The susceptibility results of Vitek 2 were exported and analyzed by WHONET 5.4 and were presented in this study. Antibiotic susceptibility pattern for imipenem and meropenem were further tested using Kirbey-bauer disk diffusion according to the last recommendation of CLSI [18]. Double-Disk Potentiation test was performed as described by Yong et al. [19], strains originally resistant to carbapenems became sensitive after the addition of Ethylene Diamine Tetra Acetic Acid (EDTA).

Multi-drug resistance of *A. baumannii* has in other studies been defined in several ways: strains resistance to antibiotics of three or more antibiotic classes, carbapenem resistance, or isolates sensitive to tigecycline and colistin only [3,5,11,12]. Among these, the definition the most widely used was that MDR strains were non-susceptible to ≥1 agent in ≥3 antimicrobial categories [1]. Accordingly, this definition was used for the purpose of this study [1]. The breakpoints used were according to the last recommendation of CLSI [18].

### 2.4. Molecular Analysis

Bacterial DNA was extracted from isolates using DNeasy Blood and Tissue kit (Qiagen, Germany) according to the manufacturer’s instructions. Carbapenemase resistance genes (*bla_KPC_*, *bla_OXA-23_*, *bla_VIM_*, and *bla_IMP_*) were detected by uniplex PCR using specific primers as described previously [20,21,22]. *A. baumannii* NCTC 13305 strain was used as positive control for *bla_OXA-51_*-like gene, *Escherichia coli* NCTC 13451 was used as negative control for *bla_OXA-51_*-like. *Klebsiella pneumonia* NCTCC 13439 and *E. coli* NCTCC 13476 were used as positive controls for VIM and IMP genes respectively. Well-characterized strains of *K. pneumoniae* carbapenemase (KPC)-producing isolates (sequenced for the targeted genes) were used as positive controls for OXA-23 and KPC genes.

### 2.5. Statistical Analysis

Using demographic variables of year and hospital unit, incidence rates were determined to assess relative new cases of MDR isolates by year and by hospital unit, I = number of new cases/the specific hospital population per 1000. Using SPSS v20 (IBM Corp, Armonk, NY, USA), an ANOVA was used to determine if there was a measurable difference in incidence per year, and odds ratios (OR) using a 95% confidence interval were used to measure the impact of MDR isolates based on the demographic variables of gender, age, nationality, hospital unit, specimen source and year calculated by Chi-square test. (OR = A × D/B × C) and 95% CI = exp(In(OR) − 1.96 × SE to exp(IN(OR) + 1.96 × SE(In(OR). Variables were compared between MDR *A. baumannii* and non-MDR *A. baumannii* within *A. baumannii* infected population to determine association with MDR.

## 3. Results

### 3.1. A. baumannii Isolates Distribution

A total of 622 *A. baumannii* isolates were collected from January 2010 to June 2020. Identification confidence was good or excellent for 98.9% of the reported isolates. Isolate distribution indicated most isolates were from adults aged 18–60 years (54.2%), male (65.6%), and Jordanian (45.3%). The majority of the isolates were from patients in the surgical unit (37.3%), followed by the ICU and cardiac care unit (CCU) 33.3% and outpatient department 21.5%, with almost half (44.1%) being from infected wounds. The frequency of isolates reported according to year was not statistically significant, although there was a slight increase in 2015 to 20.3% (Table 1).

### 3.2. A. baumannii MDR Isolates Distribution

*A. baumannii* MDR isolates resistant to antibiotics of three or more antibiotic classes was further analyzed. The total number of MDR isolates was 478 from the 622, indicating 76.8% of *A. baumannii* isolates were MDR. Several risk factors were associated with multi-drug-resistant *A. baumannii*; these included: odds of having an MDR isolate is 4 times higher in adults over 60 (OR = 4.14, CI 2.5190–6.8102, *p* < 0.0001), while children under 18 years old were 68% less likely to be multi-drug resistant (OR = 0.3095, CI 0.1828–0.5240, *p* < 0.0001). Regarding gender, the odds of having MDR was higher for men, being 2.45 times more likely than for women (OR = 2.45, CI 1.67–3.59, *p* < 0.0001). Regarding nationality, MDR isolate was about 40% less likely among Jordanian patients than those of other regional nationalities (OR = 0.5855, CI 0.4022–0.8523, *p* = 0.0052). Odds of having an MDR isolate was about 9 times more likely in inpatients compared to outpatients (CI 5.7916–13.6947, *p* < 0.0001), 10 times more likely among CCU/ICU patients compared to other units (CI 8.2338–12.1148, *p* < 0.0001), and 7× more likely among pus wounds than other specimens collected (OR = 7.03, CI 4.2416–11.658, *p* < 0.0001). Finally, there was no statistically significant risk associated with the years (*p* = 0.3933) (Table 2).

### 3.3. Incidence Rate of A. baumannii Infections

By reviewing the number of admissions per year and per unit, the incidence rates of *A. baumannii* and MDR *A. baumannii* infections were calculated. As shown in Table 3, the incidence rate of *A. baumannii* and *A. baumannii* MDR infections fluctuated slightly over the years with rates from 2010 to 2016 varying from 3.75/1000 to 4.46/1000, but there was no statistically significant increase in incidence over years (*p* = 0.3933). However, there were statistically significant differences in MDR resistance based on medical unit. Clearly, the incidence rate of *A. baumannii* infections in the surgical units was higher than most of the other hospital units at 4.75 per 1000 admissions, compared to less than 1 per 1000 in the medical and pediatric units. But the highest rates of incidence of *A. baumannii* MDR infections were in the ICU/CCU with a cumulative incidence of 23.38 per 1000 admissions, and an individual yearly incidence that ranged from 14.79–33.54/1000, over the 5-year period.

### 3.4. Antibiotic Susceptibility Pattern for A. baumannii Isolates

The antibiotic resistance rate of *A. baumannii* for the aminoglycosides antibiotic group ranged from 37.1% for amikacin, 37.2% for tobramycin to 62.6% for gentamicin (see Table 4). For the Carbapenems, the resistance rate was 68.9% for imipenem and 75.1% for meropenem. Among the tetracyclines group the resistance rate was low for tigecycline at 7.2% and minocycline at 23.8% but higher for tetracycline at 65.4%. Several groups of drugs were not effective against *A. baumannii* this included the cephalosporin group with a relatively high resistance rate at 83.5% for ceftriaxone, 99.5% for cefazolin, 99.5% for cefoxitin, 97.7% for cefuroxime, 77.9% for ceftazidime, 81.9% for cefepime, and 85.3% for cefotaxime. Fluoroquinolones were also mostly ineffective with resistance rates of 83% for ciprofloxacin and 62.5% for levofloxacin. For ampicillin, the resistance rate was 88.2%, for aztreonam it was 95.5% and for trimethoprim/sulfamethoxazole was 49.9%. There were mixed levels of resistance for Beta-lactam Inhibitors; ampicillin/sulbactam (60%) and piperacillin/tazobactam (80.3%). For colistin, the resistance rate was 2.3%. Accordingly, the lowest resistance rate was for colistin and tigecycline.

### 3.5. A. baumannii Resistance Phenotype

For each class of drug tested by the Vitek 2 System, the Antimicrobial Susceptibility Testing (AES) attempted to determine a phenotype for the strain. This was carried out by comparing the measured Minimum Inhibitory Concentrations (MICs) of the drugs within a class to a range of MICs in the AES database for strains possessing documented phenotypes. Most strains identified in this study had the following Aminoglycoside-resistant phenotypes: GEN NET AMI TOB at 33.8% (*n* = 210), GEN TOB AMI at 33.8% (*n* = 210) and TOB GEN NET at 31.5% (*n* = 196). The frequency and number of isolates that were used to determine the resistance phenotype for other antibiotic classes were too low (data not shown). Carbapenem resistance was detected in the majority of confirmed *A. baumannii* isolates. 100% of confirmed *A. baumannii* were resistant to imipenem, and 99.2% to meropenem. The Double-Disk Potentiation test with EDTA was positive in 97.7% with imipenem and in 99.2% with meropenem (Table 5).

### 3.6. Detection of Carbapenem Resistance Genes

The presence of carbapenemase genes including KPC, VIM, IMP, OXA-23, and OXA-51 were investigated in all confirmed *A. baumannii* strains using uniplex PCR. The *bla_OXA-51_* gene was found in all *A. baumannii* isolates (100%). The OXA-23 gene was found in 98.5%, VIM gene was detected in 26.6%, IMP gene was detected in 0.8%, whereas none of the confirmed *A. baumannii* isolates had the KPC gene (0%) (Table 6 and Appendix A).

## 4. Discussion

*A. baumannii* has become a common cause of infections associated with high mortality and morbidity, and typically more prevalent in those that are immunocompromised, in intensive care units and from war wounds [1,2,23]. Furthermore, *A. baumannii* has become a multi-drug-resistant microorganism worldwide [5], including resistance to many last resort options leading to great concern in the medical community [24,25,26,27,28]. Pandrug-resistant *A. baumannii* infections are increasingly being reported worldwide and are associated with high mortality [29]. Treatment options for *A. baumannii* MDR are very limited and the evidence to guide treatment is lacking with combination therapy being recommended [30,31].

As addressed earlier, there has been limited data available on incidence, risk factors, antibiotic susceptibility pattern and the genetic background of *A. baumannii* infections in both Jordan and the Middle East [8,9,10,13]. Recent conflicts in the Middle East, resulting in a large number of immigrants and refugees, and poor availability of appropriate medical care are potential contributing factors to multi-drug-resistant organisms, enhanced resistance patterns, and rising incidence rates [32,33,34]. Therefore, continued monitoring and surveillance is critical to forecast and mandate appropriate measures as they become necessary [34,35].

The findings from this study showed that most isolates were from males, aged 18-60 years, Jordanian, isolated from infected wounds, and from patients in surgery, ICU and CCU departments, which was similar to previous Jordanian studies that also showed an increased incidence of *A. baumannii* infections in adult males, ICU patients and from infected wound [12,14,15]. In addition, the *A. baumannii* MDR rate was similar to previous studies in Jordan which had reported MDR rates of 65.7% and 77.6% [11,12], a range which covers the 70% of *A. baumannii* isolates identified as MDR in this study. Adults over 60, men, non-Jordanians, ICU/CCU patients and infected wounds were significantly associated with MDR incidence, while no statistical significance was associated with years, indicating that the rates have been rather stable over the study period. Other studies have reported higher rates of MDR *A. baumannii* ranging from 88% to 100%, however the population in these studies consisted solely of critically ill patients and/or patients on ventilation [13,14,15] This is similar to findings shown in this study, indicating that critically ill patients were 10 times more likely to have a MDR form of *A. baumannii*. Some important risk factors associated with *A. baumannii* MDR like hospitalization duration and invasive procedures were not included in this study due to missing data considering the multicentric and prolonged study period. Furthermore, proper multivariate analysis considering all important risk factors could not be performed.

The current study showed the following resistance patterns: high resistance for most cephalosporins, carbapenems, fluoroquinolones, and ampicillin; moderate resistance for trimethoprim/sulfamethoxazole and ampicillin/sulbactam; low resistance rates for aminoglycosides and tetracyclines; and the lowest resistance rates for colistin and tigecycline. Antibiotic susceptibility patterns were similar to other studies in Jordan, with the lowest resistance rate being for colistin among all studies [11,14,15,36].

In this study, using data available on Vitek 2 Compact System and patient records from 2010 to 2020, 622 *A. baumannii* isolates were reported with about a 99% high confidence rate. Few samples identified by Vitek 2 and/or routine diagnostic microbiology as *A. baumannii* were *bla_OXA-51_* negative and were excluded from analysis. OXA-51 is considered intrinsic and universal in *A. baumannii* [17]. Multiple studies have concluded the usefulness, accuracy, and the timely output of the automated Vitek 2 System in the performance of antibiotic susceptibility testing in general and specifically for *A. baumannii* [37,38,39,40]. However, the ability of the Vitek 2 system to appropriately identify the accurate resistance rate for some antibiotics like tigecycline, minocycline, and colistin is debatable as it might overestimate the resistance rate for these antibiotics [40,41,42]. Furthermore, discrepancies between disk diffusion test and Vitek 2 for imipenem and meropenem sensitivity can be noted. Other studies have reported similar discrepancies with Vitek 2 test having higher agreement with broth microdilution [19,39,40,41,42] Currently, the carbaNP test is recommended for testing suspected carbapenemase production among Enterobacteriaceae, however, the test showed poor sensitivity for *A. baumannii* [18]. Broth microdilution is currently recommended for testing susceptibility to colistin [18].

All examined *Acinetobacter* isolates carried *bla_OXA-51_* gene which is a confirmed intrinsic gene in *A. baumannii* [23,43]. OXA-23 was the commonest carbapenemase gene among these isolates, found in (98.5%) of isolates, in agreement with previous studies in Jordan [13,44,45] but higher than Obeidat et al. 2014 (58%) [15]. Moreover, studies in Egypt and Qatar showed that 100% of the isolates were positive for *bla**_OXA-23_* [46,47]. Al Atrouni et al. 2016 and Rafei et al. 2015 showed that (90–93%) of their isolates were positive for *bla**_OXA-23_* in Lebanon [43,48]. On the other hand, studies in Croatia and Mexico showed 0–3% positivity for *bla**_OXA-23_* indicating geographical variation in the prevalence of the *bla**_OXA-23_* globally [49,50]. Although our study showed a prevalence of 26.6% for the *bla_VIM_*, geographical variation was reported with its prevalence also, it was 100% in Uganda [51], 53% in India [52], and 7–17% in Iran [53,54]. Contrary to the findings of Robledo et al. 2010, none of our tested isolates carried *bla_KPC_* [55]. Our *bla_KPC_* findings were consistent with earlier studies [46,56,57]. *bla_IMP_* was detected in one isolate only in this study (0.8%) in agreement with previous studies [52,53,54,58], while other studies found it to be more abundant in their isolates (61%) [59].

This study provided an insight to identify recent patterns and cases of *A. baumannii* isolates in Jordan. The strengths of the study include large number of isolates (622), collected from three tertiary central hospitals in Jordan over a long period of time (10 years), supported by in-depth phenotypic and genotypic analysis. One current limitation to this study was that the population of this study consisted only of patients seeking treatment for infections at the hospital and who had swabs sent to the microbiology laboratory that showed positive culture for *A. baumannii*, resulting in a potential self-selecting bias. Possibilities still remain that certain patients at risk of having *A. baumannii* infections might not have presented to these tertiary hospitals, or swabs might not have been obtained or processed appropriately from some patients due to lack of adherence to hospital protocol, although the hospitals included in this study follow the stewardship program and protocols requiring all patients with possible bacterial infections to have swabs sent for appropriate bacterial isolation and antibiotic susceptibility testing.

## 5. Conclusions

In conclusion, in this large, multicentric, prolonged study, we characterized over 600 *A. baumannii* isolates. Almost seventy-seven percent (76.8%) of *A. baumannii* isolates were multi-drug resistant. Elderly (60+), men, being critically ill and having infected wounds, were significantly associated with MDR isolates, with relatively stable rates over the last several years. High rates of antibiotic resistance were observed for most antibiotics, whereas the lowest resistance rates were observed for colistin and tigecycline, making them the most effective treatment options. Attention should be given to patients at higher risk for early identification of MDR *A. baumannii*. Almost all isolates were positive for the OXA-51 gene (100%) and the OXA-23 gene (98.5%), indicating the critical role of these genes. KPC and IMP genes were almost not detected (0 and 0.8% respectively). The VIM gene was detected in 26.6% of isolates for the first time in Jordan.

## Figures and Tables

**Table 1 microorganisms-10-00849-t001:** *Acinetobacter baumannii* isolates distribution (Total *n* = 622).

		Number of Isolates	Percentage %
**Age**	Children <18 years	66	10.6%
	Adults (18–60 years)	337	54.2%
	Adults above 60 years	219	35.2%
**Gender**	Male	408	65.6%
	Female	214	34.4%
**Nationality**	Jordanian	282	45.3%
	Palestinian	31	5.0%
	Libyan	107	17.2%
	Saudi	44	7.1%
	Sudanese	34	5.5%
	Syrian	23	3.7%
	Yemeni	55	8.8%
	Iraqi	26	4.2%
	Others	20	3.2%
**Department**	ICU and CCU	207	33.3%
	Outpatients	134	21.5%
	Medicine	27	4.3%
	Surgery	232	37.3%
	Pediatric and prematurity	22	3.6%
**Identification confidence by Vitek 2**	Excellent	544	87.5%
	Good	71	11.4%
	low	7	1.1%
**Specimen source**	Blood	59	9.5%
	Sputum	91	14.6%
	Urine	98	15.8%
	Wound Pus	274	44.1%
	Body fluid	25	4.0%
	Tissue	15	2.4%
	Others	60	9.6%
**Year**	2010	90	14.5%
	2011	66	10.6%
	2012	91	14.6%
	2013	75	12.1%
	2014	106	17.0%
	2015	126	20.3%

ICU: intensive care unit, CCU: cardiac care unit.

**Table 2 microorganisms-10-00849-t002:** *A. baumannii* MDR isolates distribution (Total *n* = 478).

		MDR No/Total (%)	Non-MDR No/Total (%)	OR/95% CI/*p* Value
**Age**	Children <18 (years)	36/66 (54.5)	30/66 (45.5)	OR = 0.3095, CI 0.1828–0.5240, *p* < 0.0001
	Adults 18–60 (years)	244/337 (72.4)	93/337 (27.6)	
	Adults above 60 (years)	198/219 (90.4)	21/219 (9.6)	OR = 4.1418, CI 2.5190–6.8102, *p* < 0.0001
**Gender**	Male	337/408 (82.6)	71/408 (17.4)	males versus females, OR = 2.45, CI 1.67–3.59, *p* < 0.0001
	Female	141/214 (65.9)	73/214 (34.1)	
**Nationality**	Jordanian	202/282 (71.6)	80/282 (28.4)	Jordanian vs other, OR = 0.5855, CI 0.4022–0.8523, *p* = 0.0052
	Palestinian	28/31 (90.3)	3/31 (9.7)	
	Libyan	89/107 (83.2)	18/107 (16.8)	
	Saudi	34/44 (77.3)	10/44 (22.7)	
	Sudanese	23/34 (67.6)	11/34 (32.4)	
	Syrian	21/23 (91.3)	2/23 (8.7)	
	Yemeni	46/55 (83.6)	9/55 (16.4)	
	Iraqi	17/26 (65.4)	9/26 (34.6)	
	Others	18/20 (90.0)	2/20 (10)	
**Department**	ICU and CCU	189/207 (91.3)	18/207 (8.7)	CCU/ICU vs other units, OR = 9.9876, CI 8.2338–12.1148, *p* < 0.0001
	Outpatients	56/134 (41.8)	78/134 (58.2)	
	Medicine	24/27 (88.9)	3/27 (11.1)	
	Surgery	197/232 (84.9)	35/232 (15.1)	
	Pediatric and prematurity	12/22 (54.5)	10/22 (45.5)	
**Specimen source**	Blood	47/59 (79.7)	12/59 (20.3)	Pus wound had an OR = 7.0304, CI 4.2416–11.658, *p* < 0.0001
	Sputum	85/91 (93.4)	6/91 (6.6)	
	Urine	39/98 (39.8)	59/98 (60.2)	
	Wound Pus	254/274 (92.7)	20/274 (7.3)	
	Body fluid	23/25 (92.0)	2/25 (8.0)	
	Tissue	14/15 (93.3)	1/15 (6.7)	
	Others	16/60 (26.7)	44/60 (73.3)	
**Year**	2010	66/90 (73.3)	24/90 (26.7)	*p* = 0.3933
	2011	53/66 (80.3)	13/66 (19.7)	
	2012	75/91 (82.4)	16/91 (17.6)	
	2013	59/75 (78.7)	16/75 (21.3)	
	2014	87/106 (82.1)	19/106 (17.9)	
	2015	90/126 (71.4)	36/126 (28.6)	
	2016	48/68 (70.6)	20/68 (29.4)	

OR: odds ratio, CI: 95% confidence interval, ICU: intensive care unit, CCU: cardiac care unit.

**Table 3 microorganisms-10-00849-t003:** Incidence rate of *A. baumannii* and *A. baumannii* MDR infections by year and hospital department.

		No of *A. baumannii* Cases/No of Admission per Year	*A. baumannii* Isolates Rate/1000 Admissions	No of *A. baumannii* MDR Cases/No of Admission per Year	*A. baumannii* MDR Rate/1000 Admissions
**Year**	2010	90/17,587	5.11	66/17,587	3.75
	2011	66/18,191	3.62	53/18,191	2.91
	2012	91/19,856	4.58	75/19,856	3.77
	2013	75/20,752	3.61	59/20,752	2.84
	2014	106/21,324	4.97	87/21,324	4.08
	2015	126/20,174	6.24	90/20,174	4.46
**Departments**	Surgery	231/48,680	4.75	197/48,680	4.05
	ICU/CCU	207/8855	23.38	189/8855	21.34
	Pediatrics	22/16,992	1.29	12/16,992	0.71
	Medical	27/43,357	0.62	24/43,357	0.55

**Table 4 microorganisms-10-00849-t004:** Antibiotic susceptibility pattern of *A. bummanii* by Vitek 2.

Antibiotic Name	Antibiotic Subclass	%R	%I	%S
**Amikacin**	Aminoglycosides	37.1	9.7	53.2
**Gentamicin**	Aminoglycosides	62.6	7.7	29.7
**Tobramycin**	Aminoglycosides	37.2	19.1	43.8
**Imipenem**	Carbapenems	68.9	3.2	27.9
**Meropenem**	Carbapenems	75.1	5.4	19.5
**Tetracycline**	Tetracyclines	65.4	10.3	24.3
**Tigecycline**	Tetracyclines	7.2	23.8	69
**Minocycline**	Tetracyclines	23.8	20.1	56.1
**Ampicillin**	Aminopenicillins	88.2	10.9	0.9
**Ampicillin/Sulbactam**	Beta-lactam + Inhibitors	60	15.7	24.3
**Piperacillin/Tazobactam**	Beta-lactam + Inhibitors	80.3	1.3	18.3
**Cefazolin**	Cephalosporin I	99.5	0	0.5
**Cefoxitin**	Cephalosporin II	99.5	0	0.5
**Cefuroxime**	Cephalosporin II	97.7	1.5	0.8
**Ceftazidime**	Cephalosporin III	77.9	5.1	17
**Cefotaxime**	Cephalosporin III	85.3	11.6	3.2
**Ceftriaxone**	Cephalosporin III	83.5	15.4	1.2
**Cefepime**	Cephalosporin IV	81.9	2.2	15.9
**Aztreonam**	Monobactams	95.5	3.7	0.7
**Ciprofloxacin**	Fluoroquinolones	83	0.3	16.8
**Levofloxacin**	Fluoroquinolones	62.5	23.3	14.3
**Trimethoprim/Sulfamethoxazole**	Folate pathway inhibitors	49.9	0	50.1
**Colistin**	Polymyxin	2.3	0	97.7

R: resistant, I: intermediate, S: sensitive. Colistin MIC ≤ 2 μg/mL = I and MIC > 4 μg/mL = R.

**Table 5 microorganisms-10-00849-t005:** Phenotypic test for detection of carbapenemase-producing *A. baumannii*. Carbapenemase screening test was performed with disk diffusion test while carbapenemase confirmation test was performed by Double-Disk Potentiation test.

		Resistance (%)	Susceptible (%)
**Carbapenemase screening test**	imipenem	100%	0%
	meropenem	99.2%	0.8%
		**Positive (%)**	**Negative (%)**
**Carbapenemase Confirmation test**	imipenem with EDTA	97.7%	2.3%
	meropenem with EDTA	99.2%	0.8%

**Table 6 microorganisms-10-00849-t006:** Carbapenem resistance genes among *A. baumannii*.

	Positive (%)	Negative (%)
**OXA-51**	100%	0%
**OXA-23**	98.5%	1.5%
**KPC**	0%	100%
**IMP**	0.8%	99.2%
**VIM**	26.6%	73.4%

## Data Availability

Data are available upon request.

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
