# Peer review of "Multidrug-Resistant Acinetobacter baumannii in Jordan"

_microorganisms, 2022, doi:10.3390/microorganisms10050849_

Round 1

Reviewer 1 Report

Abstract

  • I suggest modifying the sentence "Significant risk factors for MDR were.." to "Among patient from whom A. baumannii was isolated risk factors for MDR were...". This should be clarified (i.e. the analysis to identify risk factors was applied to patients having A baumannii rather than to all admitted patients)

Methods

  • How were duplicate isolates (>1 isolates from the same patient) handled? How were isolates sampled? (all consecutive isolates?, random sample? convenience sample?)
  • Were all isolates and all antibiotics assessed with both Vitek 2 and disk diffusion? Please clarify. And how was discordance between the 2 methods handled?
  • Incidence was calculated " I = # new cases/the specific hospital population per 1000". Please clarify the denominator, i.e. number of new cases per 1000 admissions per year.
  • A multivariate analysis for risk factors is lacking

Results

  • "adults over 60 are about 4 times more likely to be an MDR isolate". This is a misinterpretation of odds ratio. It would be more corect to say "odds of having an MDR isolate is 4 times higher in adults over 60. The same applies to the rest of the text.
  • In Table 2 I suggest reporting the percentage of MDR isolates in each subgroup (rather than the percentage currently reported). E.g. the percentage of MDR AB was 90.4% (198/219) in patients >60.
  • Table 2: I don't understand the rationale of grouping outpatients/medicine/surgery in one group (ICU/CCU is compared to other combined). The percentage of MDR AB was 42% (56/134) in outpatients, 89% (24/27) in medical units and 85% (197/232) in surgery. Therefore, grouping all these patients together does not makes sense.
  • Table 4; Based on current CLSI guidelines a colistin MIC ≤ 2 is now interpreted as "I" rather than "S" (https://clsi.org/media/3634/mr01ed2_sample.pdf)
  • Instead of comparing the incidence of MDR AB per year/unit it would be more interesting to compare the incidence of XDR AB.

Discussion

  • "including resistance to many last resort options leading to great concern in the medical community". Consider also discussing the emerging threat of pandrug resistance in A. baumannii (e.g. PMID: 31586417, PMID: 32927013). As the authors point out colistin and tigecycline are last resort treatment options for many A. baumannii isolates. However, resistance to these last resort antimicrobials can emerge in vivo (reviewed in PMID: 32599229, and PMID: 32677578). Treatment options for these pathogens remain very limited and the evidence to guide their treatment is lacking (reviewed in PMID: 32875545, PMID: 34827282 and PMID: 34982411)
  • Broth microdilution is currently recommended for evaluation of susceptibility to polymyxins. Very major errors are common with Vitek 2. This should be acknowledged as a limitation 
  • Some important risk factors for MDR A. baumannii isolation were not available (e.g. duration of hospitalization, invasive procedures). This should be acknowledged as a limitation
  • A multivariate analysis is lacking. Therefore, interprentation (based on univariate analysis) of higher MDR percentages in certain groups should be cautious.
  • What is the authors interprenations of higher MDR risk in Jordanians?
  • The study's population (patients with A baumannii) is inappropriate to assess risk factors for MDR AB in the overal patient population. 

Author Response

Reviewer 1:

Abstract

I suggest modifying the sentence "Significant risk factors for MDR were.." to "Among patient from whom A. baumannii was isolated risk factors for MDR were...". This should be clarified (i.e. the analysis to identify risk factors was applied to patients having A baumannii rather than to all admitted patients)

Response: thank you. The sentence in abstract, page 1, line 22 was modified as requested to read “Among patient from whom A. baumannii was isolated, assocaited risk factors for MDR were..”

Methods

How were duplicate isolates (>1 isolates from the same patient) handled?

Response: duplicate isolates (serial isolates of the same pathogen from the same patient) were excluded from the study, while more than one isolate with different phenotype and antibiotic resistance pattern were included.

The following text was added to methods section, page 2, line 86 “(consecutive and non-duplicated)”.

How were isolates sampled? (all consecutive isolates?, random sample? convenience sample?)

Response: thank you. All patients identified as A. bumannii were included in the study.

The following text was added to methods section, page 2, line 86 “(consecutive and non-duplicated)”.

Were all isolates and all antibiotics assessed with both Vitek 2 and disk diffusion? Please clarify. And how was discordance between the 2 methods handled?

Response: thank you. All isolates were tested by Vitek 2 and results were included in this study to determine the sensitivity pattern of isolates (Table 4). Kirbey-bauer disk diffusion test was used for imipenem and meropenem only and with Double-Disk Potentiation test (Table 5). Carbapenems resistance is critical to the study aims and thus was tested by 2 methods.

This was clarified in methods section, page 3, line 110-112 “The susceptibility results of Vitek 2 were exported and analyzed by WHONET 5.4 and were presented in this study. Antibiotic sensitivity patterns for imipenem and meropenem were further tested using Kirbey-bauer disk diffusion according to the last recommendation of CLSI”.

The discordance between the 2 methods were discussed in discussion section, page 11, lines 293-298 as follow “Furthermore, discrepancies between disk diffusion test and Vitek 2 for imipenem and meropenem sensitivity can be noted. Other studies have reported similar discrepancies with Vitek 2 test having higher agreement with broth microdilution [19,39-42] Currently, carbaNP test is recommended for testing suspected carbapenemase production among Enterobacteriaceae, however, the test showed poor sensitivity for A. baumannii [18].” 

Incidence was calculated " I = # new cases/the specific hospital population per 1000". Please clarify the denominator, i.e. number of new cases per 1000 admissions per year.

Response: thank you. The dominator numbers were added as requested as new columns in Table 3 (no of A. baumaannii cases/no of total admissions per year and no of A. baumannii MDR cases/ no of total admission per year).

A multivariate analysis for risk factors is lacking

Response: thank you. We agree that multivariate analysis would be more appropriate, however, due to missing data on some important risk factors (invasive procedures, hospitalization and infection duration, chronic diseases, smoking, etc) a proper multivariate analysis could not be performed. This limitation was acknowledged in discussion section page 9, line 276-277 as follow “Furthermore, proper multivariate analysis considering all risk factors could not be performed.”

Results

"adults over 60 are about 4 times more likely to be an MDR isolate". This is a misinterpretation of odds ratio. It would be more corect to say "odds of having an MDR isolate is 4 times higher in adults over 60. The same applies to the rest of the text.

Response: thank you, this was modified in result section, page 5, line 162-163. The same was applied to the text on odds ratio, page 5, lines 163-173.

In Table 2 I suggest reporting the percentage of MDR isolates in each subgroup (rather than the percentage currently reported). E.g. the percentage of MDR AB was 90.4% (198/219) in patients >60.

Response: thank you, the percentages have been reported as requested in Table 2.

Table 2: I don't understand the rationale of grouping outpatients/medicine/surgery in one group (ICU/CCU is compared to other combined). The percentage of MDR AB was 42% (56/134) in outpatients, 89% (24/27) in medical units and 85% (197/232) in surgery. Therefore, grouping all these patients together does not makes sense.

Response: ICU/CCU patients are known to be at higher risks that’s why it was compared to other units combined to allow analysis of 2x2. You could compare ICU/CCU with one other unit, but it would be hard to decide which unit is more appropriate to compare with. Comparing MDR AB in ICU/CCU to other units separately also showed significant increase in ICU/CCU, however, you come up with many OR and P values that would be difficult to follow. Similar approach was applied to other variables (ex Jordanian nationalities vs others combined, pus wound compared to other specimens combined, etc).

Table 4; Based on current CLSI guidelines a colistin MIC ≤ 2 is now interpreted as "I" rather than "S" (https://clsi.org/media/3634/mr01ed2_sample.pdf)

Response: thank you. This is added to Table 4.

Instead of comparing the incidence of MDR AB per year/unit it would be more interesting to compare the incidence of XDR AB.

Response: thank you. While it would be interesting to add the incidence of XDR AB, however, the numbers are too low to allow proper analysis and this would be a deviation of the primary aim of the manuscript focusing on MDR AB.

Discussion

"including resistance to many last resort options leading to great concern in the medical community". Consider also discussing the emerging threat of pandrug resistance in A. baumannii (e.g. PMID: 31586417, PMID: 32927013).

Response: Thank you. This was added to discussion sections as requested “Pandrug-resistant A. baumannii are increasingly being reported worldwide and are associated with high mortality [29]”

As the authors point out colistin and tigecycline are last resort treatment options for many A. baumannii isolates. However, resistance to these last resort antimicrobials can emerge in vivo (reviewed in PMID: 32599229, and PMID: 32677578).

Response: The above references were added to discussion as requested “including resistance to many last resort options leading to great concern in the medical community [24-28]”.

Treatment options for these pathogens remain very limited and the evidence to guide their treatment is lacking (reviewed in PMID: 32875545, PMID: 34827282 and PMID: 34982411)

Response: The above references (PMID: 32875545, PMID: 34827282) were added to discussion “Treatment options for A. baumannii MDR are very limited and the evidence to guide treatment is lacking with combination therapy being recommended [30,31].”

Broth microdilution is currently recommended for evaluation of susceptibility to polymyxins. Very major errors are common with Vitek 2. This should be acknowledged as a limitation

Response: thank you, this limitation was acknowledges in discussion, page 10, lines 298-299. Furthermore, the following text was added to lines 294-295 “Broth microdilution is currently recommended for testing susceptibility to colistin [18].”

Some important risk factors for MDR A. baumannii isolation were not available (e.g. duration of hospitalization, invasive procedures). This should be acknowledged as a limitation.

Response: thank you. This was acknowledged as a limitation in discussion section, page 9, line 273-276 as follow “Some important risk factors associated with A. baumannii MDR like hospitalization duration and invasive procedures were not included in this study due to missing data considering the multicentric and prolonged study period.”

A multivariate analysis is lacking. Therefore, interprentation (based on univariate analysis) of higher MDR percentages in certain groups should be cautious.

What is the authors interprenations of higher MDR risk in Jordanians?

The study's population (patients with A baumannii) is inappropriate to assess risk factors for MDR AB in the overal patient population.

Response: thank you. The limitation of not including a multivariate analysis due to absence of important risk factors, and population of the study have been acknowledged in discussion, page 10, lines 276-277 “Furthermore, proper multivariate analysis considering all important risk factors could not be performed.”

Reviewer 2 Report

Comments
Line 36-37: Should be "bacterium" (verb is singular, not plural). Bacillus is a noun,  not an adjective ("rod-shaped" is).
Line 50: In this and some other places, should be "incidence of A. baumannii infection".
Line 58: "While the most effective..."  is not a sentence.
Figure 1: I am confused; shouldn't oxa51 be positive in all isolates? Patients 10 and 14 do not have convincing bands, yet they seem to be counted as positive. 

In general, the English is sufficient for clarity, but there are various typos, misused words, and ungrammatical phrases scattered throughout that ideally should be improved prior to publication.

The value of this article might be enhanced by providing more data related to demographic shifts (movement of refugees etc. as hinted at in the article). 

Author Response

Reviewer 2:

Comments

Line 36-37: Should be "bacterium" (verb is singular, not plural). Bacillus is a noun, not an adjective ("rod-shaped" is).

Response: thank you. This was corrected.

Line 50: In this and some other places, should be "incidence of A. baumannii infection".

Response: thank you. This was corrected in line 50 and all over the manuscript.

Line 58: "While the most effective..."  is not a sentence.

Response: thank you. This is corrected now to read “The most effective drugs were imipenem and meropenem against A. baumannii…”.

Figure 1: I am confused; shouldn't oxa51 be positive in all isolates? Patients 10 and 14 do not have convincing bands, yet they seem to be counted as positive.

Response: that is correct. Despite the high accuracy of Vitek 2 in isolates identification (99%) some samples were OXA-51 negative. These samples were excluded from the study as its generally agreed that OXA-51 is intrinsic to A. baumannii.

This was further clarified in methods section, page 3, lines 104-105 “Samples that were blaOXA-51 negative were excluded from the study.” and in discussion section, page 10, lines 286-288 “Few samples identified by Vitek 2 and/or routine diagnostic microbiology as A. bau-mannii were blaOXA-51 negative and were exclude from analysis. OXA-51 is considered intrinsic and universal in A. baumannii [17]”

In general, the English is sufficient for clarity, but there are various typos, misused words, and ungrammatical phrases scattered throughout that ideally should be improved prior to publication.

Response: Thank you. English was revised by a native speaker Dr Adam Essa.

The value of this article might be enhanced by providing more data related to demographic shifts (movement of refugees etc. as hinted at in the article).

Response: thank you, while this data would enhance the value of the article, it is not accessible to the study authors and could not be included.

Reviewer 3 Report

The presented paper gives interesting new insights about the occurence of A. baumannii in patients from 3 hospitals in Amman, Jordan. The findings regarding antibiotic resistence patterns are in line with previous findings from the Middle East region confirming colistin and tigecycline being the antibiotics with highest sensitivity rates (mordern drugs i.e cefiderocol or combinations i.e. ceftazidiem/avibactame were not included in the study design). Based on the study results recomendations for the antibiotic use might be adjusted. Therefore, the significance of the content is high.

However, I would recommend revision of the manuscript by a native speaker to improve the overall quality.

Further, I would recommend to add the paper of Higgins et al. (Antibiotics 2021, 10, 291. https://doi.org/10.3390/antibiotics10030291) to the references and the cite it at line 77.

Please explain in the 2.4 section (line 129): what means "well characterized strains of K. pneumoniae"..?

Section 3.5 (line 216): It is not obvious what means "of those tested"

Line 306 (Conclusion): the first sentence doesn't make sense

Following typos:

  • strain names have to be written in italics (i.e. A. calcoaceticus in line 42 and others)
  • line 37: "incidence" is not the convenient expression of an occurrence of bacteria in patients
  • line 102 / 109: BioMérieux
  • line 126: (19-21) in parentheses
  • line 159: "adults ... to be a MDR isolate" does not make sense
  • line 179: please write "statistically"
  • line 198: Fluoroquinolones (at beginning of the sentence)
  • line 288: Missing a dot after (49)

Author Response

Reviewer 3:

The presented paper gives interesting new insights about the occurence of A. baumannii in patients from 3 hospitals in Amman, Jordan. The findings regarding antibiotic resistence patterns are in line with previous findings from the Middle East region confirming colistin and tigecycline being the antibiotics with highest sensitivity rates (mordern drugs i.e cefiderocol or combinations i.e. ceftazidiem/avibactame were not included in the study design). Based on the study results recomendations for the antibiotic use might be adjusted. Therefore, the significance of the content is high.

Response: thank you. No further action needed.

However, I would recommend revision of the manuscript by a native speaker to improve the overall quality.

Response: thank you. English was revised by a native speaker Dr Adam Essa.

Further, I would recommend to add the paper of Higgins et al. (Antibiotics 2021, 10, 291. https://doi.org/10.3390/antibiotics10030291) to the references and the cite it at line 77.

Response: thank you. The suggested reference was cited and added to the references list.

Please explain in the 2.4 section (line 129): what means "well characterized strains of K. pneumoniae"..?

Response: these strains were extensively characterized by molecular and phenotypic methods and were sequenced for the targeted genes (OXA-23 and KPC genes) as part of another ongoing study.

The following text was modified in methods section, page 3, lines 132-133 to read” Well characterized strains of K. pneumoniae carbapenemase (KPC)-producing isolates (sequenced for the targeted genes) were used as positive controls for OXA-23 and KPC genes.”

Section 3.5 (line 216): It is not obvious what means "of those tested"

Response: this was clarified as “of confirmed A. baumannii”.

Line 306 (Conclusion): the first sentence doesn't make sense

Response: the sentence was corrected to read “in this large, multicentric, prolonged study we characterize over 600 A. baumannii isolates”.

Following typos:

  • strain names have to be written in italics (i.e. A. calcoaceticus in line 42 and others)
  • line 37: "incidence" is not the convenient expression of an occurrence of bacteria in patients
  • line 102 / 109: BioMérieux
  • line 126: (19-21) in parentheses
  • line 159: "adults ... to be a MDR isolate" does not make sense
  • line 179: please write "statistically"
  • line 198: Fluoroquinolones (at beginning of the sentence)
  • line 288: Missing a dot after (49)

Response: thank you. All above mentioned typos were corrected.

Reviewer 4 Report

  1. Please use italics for all names of microorganisms (gender and species) (eg. Line 38)
  2. English revising is suggested
  3. Line 58-59: These two sentences could be merged. In that case, delete ‘however’
  4. The word ‘antimicrobials’ sounds better than the word ‘antibiotics’
  5. Line 125: unipelex or uniplex?
  6. Please mention the details of the SPSS that are commonly mentioned when citing the software in papers (for example, manufacturer, country etc.)
  7. Line 137. I don’t fully understand how OR was calculated. What type of statistical test was used in this instance? Was it a Chi-square test?
  8. Line 147: Please define CCU, as it is mentioned in the text for the first time
  9. Line 149: Do you mean that there was no statistically significant difference?
  10. Table 1. ICU: intensive care unit, CCU: 151 cardiac care unit should be moved to a footnote
  11. Line 155: do you mean ‘resistant’?
  12. Table 2. OR: odds ratio, CI: 95% confidence 171 interval, ICU: intensive care unit, CCU: cardiac care unit should be moved to a footnote
  13. Since it is not clear what the statistics at the last column suggest, I would change the table to have one column with the # (%) of the MDR, one column with the # (%) of the non-MDR (I guess the statistics should compare these two columns, since simply comparing for example males vs females is probably wrong, as the main population in Table 1, from where the MDR patients are a subset, also have a male predominance) and one column with the statistics comparing the previous two columns
  14. Line 255. I agree that being male, over 60 years old etc, means that you are more likely to acquire baumannii, and probably to acquire an MDR A. baumannii, however, the question who is more likely to acquire an MDR A. baumannii among those that have an A. baumannii infection is not answered. The statistical analysis mentioned in the previous comment is more appropriate for that
  15. Line 256. I am not sure that the statistical analysis in this manuscript is designed to identify independent risk factors. It may identify an association, but not causality and independent risk factors
  16. Line 86 says that the study period was 2010-2020, while line 269 says 2010-2016. Which of the two is correct?
  17. I am not sure that showing Figure 1 is necessary. It could be moved to supplementary materials

Author Response

Reviewer 4:

  • Please use italics for all names of microorganisms (gender and species) (eg. Line 38)

Response: thank you. This was corrected in line 38 and through the manuscript.

  • English revising is suggested

Response: thank you. English was revised by a native speaker Dr Adam Essa

  • Line 58-59: These two sentences could be merged. In that case, delete ‘however’

Response: the first sentence was modified.

  • The word ‘antimicrobials’ sounds better than the word ‘antibiotics’

Response: “antibiotics” was replaced with “antimicrobials” wherever appropriate.

  • Line 125: unipelex or uniplex?

Response: this was corrected to “uniplex”.

  • Please mention the details of the SPSS that are commonly mentioned when citing the software in papers (for example, manufacturer, country etc.)

Response: SPSS details were mentioned as suggested “(IBM Corp, Armonk, NY, USA)”

  • Line 137. I don’t fully understand how OR was calculated. What type of statistical test was used in this instance? Was it a Chi-square test?

Response: Variables were compared between MDR and non-MDR within A. baumannii infected population to determine association with MDR. For example, with gender variable the table looks like:

MDR A. baumannii

Non-MDR A. baumannii

Male

337

71

Female

141

73

Total

478

144

OR can be calculated manually using a simple equation as mentioned in the statistical analysis section “(OR = AxD/BxC” = 337X73/141X71=2.45). Furthermore, OR, 95% CI, and P value using Chi-square test were calculated using SPSS. The same was applied for other variables.

The following text was added to statistical analysis section, page 3, line 142 “calculated by Chi-square test. (OR = AxD/BxC) and 95% CI = exp(In(OR)-1.96 xSE to exp(IN(OR) + 1.96 x SE(In(OR).

New column was added to Table 2 simailr to the table above to further calrify how OR and P values were calucated.

  • Line 147: Please define CCU, as it is mentioned in the text for the first time

Response: CCU was defined in line 147 as “cardiac care unit”.

  • Line 149: Do you mean that there was no statistically significant difference?

Response: in this results section (lines 148-155) and the cited Table 1, only descriptive analysis was reported. Inferential statistical analysis could be performed as no control group co compare with.

  • Table 1. ICU: intensive care unit, CCU: 151 cardiac care unit should be moved to a footnote

Response: ICU and CCU were moved to footnote.

  • Line 155: do you mean ‘resistant’?

Response: yes, and was corrected to read “resistant

  • Table 2. OR: odds ratio, CI: 95% confidence 171 interval, ICU: intensive care unit, CCU: cardiac care unit should be moved to a footnote

Response: they were moved to footnote.

  • Since it is not clear what the statistics at the last column suggest, I would change the table to have one column with the # (%) of the MDR, one column with the # (%) of the non-MDR (I guess the statistics should compare these two columns, since simply comparing for example males vs females is probably wrong, as the main population in Table 1, from where the MDR patients are a subset, also have a male predominance) and one column with the statistics comparing the previous two columns

Response: thank you. This is a good suggestion to clarify Table 2. Two columns were added as suggested (MDR No/total (%) and non-MDR no/total (%)). The statistical column did not change as it describes exactly what the respectful reviewer expected/requested (the stat compare the two columns of MDR versus non-MDR in each variable). In fact, odds ratio, Chi-square test, and P values cannot be calculated without having 2x2 cross table.

The following text was added to statistical analysis section, page 3, lines 143-145 “Variables were compared between MDR A. baumannii and non-MDR A. baumannii within A. baumannii infected population to determine association with MDR.”

  • Line 255. I agree that being male, over 60 years old etc, means that you are more likely to acquire baumannii, and probably to acquire an MDR  baumannii, however, the question who is more likely to acquire an MDR A. baumannii among those that have an A. baumannii infection is not answered. The statistical analysis mentioned in the previous comment is more appropriate for that

Response: the statistical analysis mentioned in the previous comment was applied. Variables were compared between MDR and non-MDR within A. baumannii infected population to determine association with MDR. Please see comments to points 7 and 13 above.

  • Line 256. I am not sure that the statistical analysis in this manuscript is designed to identify independent risk factors. It may identify an association, but not causality and independent risk factors

Response: thank you. The wording was modified from “independent risk factors” to “associated risk factors” or “significant association” through the whole manuscript.

  • Line 86 says that the study period was 2010-2020, while line 269 says 2010-2016. Which of the two is correct?

Response: was corrected to “2010-2020”.

  • I am not sure that showing Figure 1 is necessary. It could be moved to supplementary materials

Response: Figure 1 was moved to supplementary materials. 

Round 2

Reviewer 1 Report

manuscript appropriately revised

Author Response

Comments and Suggestions for Authors

manuscript appropriately revised.

Thank you no further action required. 

Reviewer 4 Report

The manuscript has been significantly improved. I have only some minor comments:

  1. Figure 1 (which is to be moved to the supplementary materials) can still be seen in the manuscript. It can be deleted.
  2. There is a symbol in the upper-left cell of Table 5. It can be deleted.

Author Response

The manuscript has been significantly improved. I have only some minor comments:

  1. Figure 1 (which is to be moved to the supplementary materials) can still be seen in the manuscript. It can be deleted.

Response: thank you, Figure 1 is deleted. 

  1. There is a symbol in the upper-left cell of Table 5. It can be deleted.

Response: thank you. the symbol is deleted.